# Graph Neural Networks for Soft Semi-Supervised Learning on Hypergraphs

## Abstract

Graph-based semi-supervised learning (SSL) assigns labels to initially unlabelled vertices in a graph. Graph neural networks (GNNs), esp. graph convolutional networks (GCNs), inspired the current-state-of-the art models for graph-based SSL problems. GCNs inherently assume that the labels of interest are numerical or categorical variables. However, in many real-world applications such as co-authorship networks, recommendation networks, etc., vertex labels can be naturally represented by probability distributions or histograms. Moreover, real-world network datasets have complex relationships going beyond pairwise associations. These relationships can be modelled naturally and flexibly by hypergraphs. In this paper, we explore GNNs for graph-based SSL of histograms. Motivated by complex relationships (those going beyond pairwise) in real-world networks, we propose a novel method for directed hypergraphs. Our work builds upon existing works on graph-based SSL of histograms derived from the theory of optimal transportation. A key contribution of this paper is to establish generalisation error bounds for a one-layer GNN within the framework of algorithmic stability. We also demonstrate our proposed methods' effectiveness through detailed experimentation on real-world data. We have made the code available.

## 1 Introduction

In the last decade, deep learning models have been successfully embraced in many different fields and proved to achieve unprecedented performance on a vast range of applications Krizhevsky et al. (2012); Goodfellow et al. (2014); Bahdanau et al. (2015); LeCun et al. (2015). Graph Convolutional Network (GCN) Kipf & Welling (2017) was recently proposed as an adaptation of a particular deep learning model (i.e., convolutional neural networks Lecun et al. (1998)) to enable handling graph-structured data. GCN was shown to be, in particular, effective in semi-supervised learning on attributed graphs. GCNs have inspired the current state-of-the art models for graph-based SSL Wu et al. (2019a); Veličković et al. (2018); Vashishth et al. (2019). GCNs inherently assume that the labels of interest are numerical or categorical variables.

However, in many real-world applications such as co-authorship networks, recommendation networks, etc., vertex labels can be naturally represented by probability distributions or histograms. Moreover, these real-world network datasets have complex relationships going beyond pairwise associations. Such relationships can be modelled naturally and flexibly by hypergraphs. Moreover, hypergraphs can encode additional relationships with directions as illustrated in Figure 1 and these hypergraphs are directed hypergraphs Gallo et al. (1993).

Inspired by a prior work Solomon et al. (2014) that generalised label propagation to graph-based soft SSL setting and motivated by the fact that GNNs have inspired state-of-the-art models for traditional graph-based SSL, we make the following contributions.

- We explore GNNs for soft SSL in which vertex labels are probability distributions. Motivated by real-world applications, we propose DHN (Directed Hypergraph Network), a novel method for directed hypergraphs. DHN can be applied for soft-SSL using existing tools from optimal transportation (Section 3).
- We provide generalisation error bounds for a one-layer GNN within the framework of algorithmic stability. We establish that such models, which use filters with bounded eigenvalues

**co-authorship network**

**recommendation network**

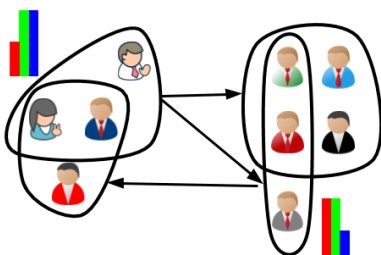
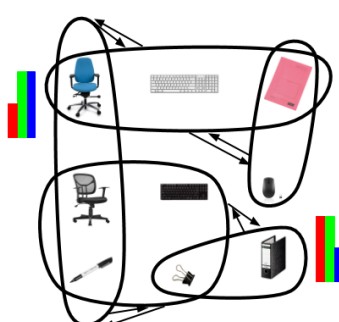

Figure 1: (Best seen in colour) Examples of real-world networks modelled as directed hypergraphs in which labels on the vertices can be represented by probability distributions. To the left is a co-authorship network in which vertices are authors, hyperedges are collaborations (documents), and directions are citations between documents. Research topic interests of authors can be naturally represented by probability distributions (labels are shown for a couple of authors in the figure). To the right is a recommendation network in which vertices are products, hyperedges are users and contain all the products bought by them, and directions represent user similarity (two-way). Product ratings can be naturally represented by probability distributions (labels are shown for a couple of products in the figure). Please see Section 1 for more details.

    independent of graph size, can satisfy the strong notion of uniform stability and thus is generalisable. In particular, the algorithmic stability of a one-layer GNN depends on the largest absolute eigenvalue of the graph convolution filter (Section 4).

- We demonstrate DHN's effectiveness through detailed experimentation on real-world data. In particular, we demonstrate superiority over state-of-the-art hypergraph-based neural networks. We provide new empirical benchmarks for soft-SSL on directed hypergraphs and make the code available to foster reproducible research (Section 5).

## 2 RELATED WORK

**Graph-based deep learning:** *Geometric deep learning* Bronstein et al. (2017) is an umbrella phrase for emerging techniques attempting to generalise (structured) deep neural networks to non-Euclidean domains such as graphs and manifolds. GCN Kipf & Welling (2017) and their various extensions are the current state-of-the art for graph-based SSL Wu et al. (2019a); Veličković et al. (2018); Monti et al. (2018); Vashishth et al. (2019); Ma et al. (2019); Qu et al. (2019) and graph-based unsupervised learning Hamilton et al. (2017a); Veličković et al. (2019) problems. GCNs have also been applied for semi-supervised graph classification Li et al. (2019).

The reader is referred to a comprehensive literature review Bronstein et al. (2017) and extensive surveys Hamilton et al. (2017b); Battaglia et al. (2018); Zhang et al. (2018); Wu et al. (2019b); Sun et al. (2018); Zhou et al. (2018) on this topic. Recently, graph-based deep models (also message-passing neural networks Gilmer et al. (2017)) have been analysed theoretically Dehmamy et al. (2019); Ying et al. (2019); Knyazev et al. (2019); Chen et al. (2019b); Maron et al. (2019); Morris et al. (2019); Xu et al. (2019b); Chen et al. (2019a); Kawamoto (2018); Xu et al. (2018); Chen et al. (2018). We note a work on stability and generalisation bounds of GCNs Verma & Zhang (2019).

**Learning on hypergraphs:** Hypergraph is a combinatorial structure consisting of vertices and hyperedges, where each hyperedge is allowed to connect any number of vertices, thus generalizing graphs. This additional flexibility facilitates the capture of higher order interactions among objects; applications have been found in many fields such as computer vision Govindu (2005), network clustering Demir et al. (2008), folksonomies Ghoshal et al. (2009), cellular networks Klamt et al. (2009), and community detection Chien et al. (2018).

The seminal work on hypergraphs Zhou et al. (2007) introduced the popular Agarwal et al. (2006); Feng et al. (2018; 2019) clique expansion of a hypergraph. Hypergraph neural networks (HGNN) Feng et al. (2019) use the clique expansion while HyperGCN Yadati et al. (2019) uses the mediator-

based Laplacian to extend GCNs to hypergraphs. Another line of work uses the mathematically appealing tensor methods Shashua et al. (2006); Bulò & Pelillo (2009); Kolda & Bader (2009) but they are limited to uniform hypergraphs. Recent developments work for arbitrary hypergraphs and fully exploit the hypergraph structure Hein et al. (2013); Zhang et al. (2017); Chan & Liang (2018); Wendler et al. (2019). These developments are motivated from the spectral theory of hypergraphs which is an active area of research Li & Milenkovic (2017); Chien et al. (2019); Chitra & Raphael (2019); Li & Milenkovic (2018b); Li et al. (2018a); Li & Milenkovic (2018a)

**Graph-based soft SSL:** Researchers have shown that using unlabelled data during training can improve label prediction significantly Chapelle et al. (2010); Zhu et al. (2009); Subramanya & Talukdar (2014); Yang et al. (2016). While most methods assume that labels of interest are numerical or categorical variables, other works "soften" this assumption and handle "soft labels" such as histograms Corduneanu & Jaakkola (2005); Tsuda (2005). One way of propagating histograms is to minimise the Kullback-Leibler (KL) divergence Subramanya & Bilmes (2011). Recent studies have replaced the metric-agnostic KL divergence with metric-aware Wasserstein distance (interactions between histogram bins) for graphs Solomon et al. (2014) and hypergraphs Gao et al. (2019).

**Embeddings in Wasserstein space:** There exist at least a couple of recent works that embed Gaussian distributions in the Wasserstein space Muzellec & Cuturi (2018); Zhu et al. (2018). Inspired by a recent work Frogner et al. (2019), in this work, we focus on embedding input data as a discrete probability distrirbution on a fixed support set. The Wasserstein distance and its gradient require the solution of a linear program Villani (2008) and are costly to compute Peyré & Cuturi (2019). A popular efficient approximation is the Sinkhorn divergence Cuturi (2013) in which the underlying problem is regularised and is computed efficiently by a fixed-point iteration. Recent works have shown that it is suitable for gradient-based optimisation through automatic differentiation Frogner et al. (2015); Genevay et al. (2018); Frogner et al. (2019). A couple of recent works compute Wasserstein distance between graph pairs Xu et al. (2019a); Vayer et al. (2019).

## 3 METHOD

In this section, we first describe soft SSL on directed hypergraphs and then propose DHN (Directed Hypergraph Network) for the problem.

### 3.1 DIRECTED HYPERGRAPH

A directed hypergraph Gallo et al. (1993) is an ordered pair $\mathcal{H} = (V, E_d)$ where $V = \{v_1, \cdots, v_n\}$ is a set of $n$ vertices and

$$E_d = \{(t_1, h_1), \cdots, (t_m, h_m)\} \subseteq 2^V \times 2^V$$

is a set of $m$ directed hyperedges. Each element in $E_d$ is an ordered pair $(t, h)$ where $t \subseteq V$ is the *tail* and $h \subseteq V$ is the *head* with $t \neq \emptyset$, $h \neq \emptyset$. Denote the set of all undirected hyperedges by $E$.

$$E = \bigcup_{(t,h) \in E_d} \left( t \cup h \right).$$

Denote $I \in \{0, 1\}^{|V| \times |E|}$ to be the incidence matrix of $E$ i.e. $I(v, e) = 1$ if $v \in e$ and 0 otherwise.

### 3.2 SOFT SSL ON DIRECTED HYPERGRAPHS

We consider the problem of predicting probability distributions for the vertices in $\mathcal{H} = (V, E_d)$ given a typically small subset $V_k \subseteq V$ of vertices with known distributions. In this work, we are concerned with discrete distributions modelled on a metric space i.e. an ordered pair $(M, C)$ in which $M$ is a set and $C$ is the cost function (metric) associated with the set. Furthermore, we assume that we are provided with a feature matrix, $X_V \in \mathbb{R}^{n \times D_V}$, in which each vertex $v \in V$ is represented by a $D_V$-dimensional feature vector $x_v$ (here $n = |V|$). We are also provided with a hyperedge feature matrix $X_E \in \mathbb{R}^{m \times D_E}$ with $x_e, e \in E$ as $D_E$-dimensional feature representations (here $m = |E_d|$).

Our objective is to learn a labelling function $Z = h\big(\mathcal{H}, X_V, X_E\big)$ that maps each vertex to a probability distribution in the space of discrete probability distributions $\mathcal{P}_F(M)$ on $F$ atoms ($F$ is number of histogram bins) defined on the metric space $(M, C)$. The cost function $C$ can be represented by a non-negative symmetric matrix of size $F \times F$. Note that each row of $Z \in [0, 1]^{n \times F}$ maps each vertex $v \in V$ to a probability distribution $Z_v \in [0, 1]^F$.

The function $h$ is going to be trained on a supervised loss, $L$ ,w.r.t to the vertices in $V_k$ so that the trained $h$ can be used to predict distributions of all the vertices in $V \setminus V_k$. We now give an example application and then the details of the labelling function $h$ followed by the supervised loss $L$.

**Example application:** Predicting topic distributions of authors in co-authorship networks can be posed as a soft SSL problem on directed hypergraphs. $V$ represents the set of authors, $E$ the set of all collaborations (documents), $E_d$ the citation relationships among the documents, $F$ the number of possible research interests of authors (Machine Learning, Theoretical Computer Science, etc.), $X_V$ and $X_E$ any available features on the authors and documents respectively (e.g. text attributes).

### 3.3 DHN (DIRECTED HYPERGRAPH NETWORK)

Hypergraphs contain hyperedges in which relationships can go beyond pairwise and hence are challenging to deal with. A flexible way to embed vertices of a hypergraph is to "approximate" the hypergraph by a suitable graph and then apply traditional graph-based methods on the vertices. Two notable candidates of $h$ are Hypergraph neural network (HGNN) Feng et al. (2019) and Hypergraph Convolutional Network (HyperGCN) Yadati et al. (2019). HGNN uses the clique expansion of the hypergraph Zhou et al. (2007) while HyperGCN uses the mediator-based Laplacian Chan & Liang (2018) to approximate the input hypergraph. However, they are restricted to undirected hyperedges and also cannot exploit the hyperedge feature matrix $X_E$.

A key idea of our approach is to treat each hyperedge $e \in E$ as a vertex of the graph $\mathcal{G} = (E, E_d)$. We then pass $\mathcal{G}$ through a graph neural network to obtain $H_E = f_{GNN}(\mathcal{G}, X_E)$ so that the initial features, $X_E$, are refined to $H_E$. We then propose the layer-wise propagation rule of $DHN$ as:

$$H_V^{(t+1)} = \sigma\left( \left[ H_V^{(t)}, \ I \cdot H_E^{(t)} \cdot \Theta^{(t)} \right] \right), \quad t = 0, \cdots, \tau - 1 \tag{1}$$

where $[\cdot, \cdot]$ denotes concatenation, $t$ is the time step, $I$ is the incidence matrix, $H_E^{(t+1)} = \sigma_1\left( I^T H_V^{(t)} \right)$ for $t = 1, \cdots, \tau - 1$, $H_E^{(0)} = f_{GNN}(\mathcal{G}, X_E)$, $\sigma$ and $\sigma_1$ are non-linear activation functions, and $\tau$ is the total number of time steps with $H_V^{(0)} = X_V$. Note that the labelling function $Z = h(\mathcal{H}, X_V, X_E) = \text{softmax}\left( H_V^\tau \right)$ where softmax is applied row-wise.

### 3.4 THE SUPERVISED LOSS $L$

A crucial observation here is that because of the softmax layer, the output of $h$ is (already) inherently a probability distribution. For each vertex $v \in V_k$, the predicted distribution $Z_v$ and the (known) true distribution $Y_v$ must be "close" to each other. A natural way to compare probability distributions is to use the KL-divergence between $Y_v$ and $Z_v$. However, KL-divergence cannot exploit the metric space $(M, C)$ and suffers from stability issues Chen et al. (2016). In this work, we use the more stable Wasserstein distance to exploit the metric space Gao et al. (2019).

$$L = \sum_{v \in V_k} W_p\left( Z_v, Y_v \right) \qquad \text{where} \quad W_p(\mu, \nu) = \left( \inf_{\pi \in \Pi(\mu, \nu)} \int_{M \times M} C(x_1, x_2)^p d\pi(x_1, x_2) \right)^{\frac{1}{p}}.\tag{2}$$

For discrete distributions, $W_p$ is the solution of a linear program. For practical purposes, we compute the regularised distance using the Sinkhorn algorithm. Please see Appendix A.1 for more details.

**Optimisation:** We call DHN optimised with the Wasserstein loss as Soft-DHN. All parameters are learned using stochastic gradient descent (SGD). Please see Appendix for time complexity.

## 4 THEORETICAL ANALYSIS: GENERALISATION ERROR BOUND

In this section, we establish generalisation error bounds for a one-layer GNN by extending the results of a traditional GCN Verma & Zhang (2019) to the soft SSL setting with Wasserstein loss. The main novelty is to generalise the error bounds to the learning problem "valued in the Wasserstein space". The main challenge is that the Wasserstein space is an abstract metric space without linear structure.

The section is organised as follows. We first introduce all the notations needed (ego-graph view, semi-supervised learning setting, etc.). We then give single layer and SGD bounds using the nota-

tions. We finally give the main result (Theorem 1) which states that a GNN trained with Wasserstein loss has the same generalisation error bound as the traditional GCN (trained with cross entropy).

Let $G = (V, E)$ be a connected graph with $|V| = n$ vertices. We consider GNN of a single layer

$$f(X, \Theta) = \sigma(KX\Theta) \tag{3}$$

where $X \in \mathbb{R}^{n \times d}$ is the feature matrix ($n$ is the number of vertices in a graph, $d$ is the dimension of the feature vectors), $K = g(L_G)$ is a graph filter (typically symmetrically normalised adjacency with self loops, and $L_G \in \mathbb{R}^{n \times n}$ is the graph Laplacian), and $\Theta \in \mathbb{R}^{d \times F}$ is the set of parameters. We note that our proposed DHN falls under this formulation in special circumstances. Specifically if the non-linearity $\sigma_1$ in Equation 1 is removed we get the kernel $K = II^T$ (also known as the clique expansion of the hypergraph Zhou et al. (2007).) The non-linearity $\sigma$ in Equation 3 is the softmax function acting on each row of the product $g(L_G)X\Theta \in \mathbb{R}^{n \times F}$; the output is of dimension $n \times F$, where each output row is a discrete probability distribution, i.e.,

$$f(X, \Theta) \geq 0 \quad \text{and} \quad f(X, \Theta)\mathbf{1}_F = \mathbf{1}_n$$

where $\mathbf{1}_F = (1, \ldots, 1)^F \in \mathbb{R}^F$, and similarly for $\mathbf{1}_n$. Without loss of generality, we assume $d = 1$. Note that in order for the output to be nontrivial probability distributions, we must assume $F > 1$.

We adopt an ego-graph view (Verma & Zhang, 2019, formula (2)) to simplify our discussion for local behavior of the soft GCN at a particular vertex. Whenever no confusion arises, we identify a vertices $x$ and $\chi$ in the graph $G$ with their respective $D$-dimensional feature vectors. Thus the output of $f$ at $x \in V$ is

$$f(x, \Theta) = \sigma\left(\sum_{\chi \in \mathcal{N}(x)} K_{x\chi}\chi\Theta\right) = \sigma\left(\left(\sum_{\chi \in \mathcal{N}(x)} K_{x\chi}\chi\right) \cdot \Theta\right)$$

where $\mathcal{N}(x)$ denotes for the one-hop neighborhood of $x$ with respect to the adjacency relation defined by matrix $K$, and $K_{x\chi} \in \mathbb{R}$ stands for the entry in $K \in \mathbb{R}^{n \times n}$ that describes the adjacency relation between vertices $x$ and $\chi$. Let $E_x := \sum_{\chi \in \mathcal{N}(x)} K_{x\chi}\chi \in \mathbb{R}$ so that $f(x, \Theta) = \sigma(E_x \cdot \Theta)$.

We consider the supervised learning setting, and learn GNN from the training set $\{z_i = (x_i, y_i), i = 1, \ldots, m\}$ sampled i.i.d. from the product space $V \times \mathcal{P}_F$ with respect to probability distribution $\mathcal{D}$ on this product space, where $\mathcal{P}_F$ is the space of discrete probability distributions on $F$ atoms. The output of softmax lies in $\mathcal{P}_F$, which is a convex cone. For any new data $z = (x, y) \sim \mathcal{D}$, we evaluate the performance of GNN $f$ using a Wasserstein cost

$$\ell(f(\cdot, \Theta), z) = \ell(f(\cdot, \Theta), (x, y)) = W(f(x, \Theta), y).$$

Here the Wasserstein cost is defined with respect to a cost function penalizing moving masses across bins. Since we are working only with histograms in GNN, we shall use a cost function $C \in \mathbb{R}^{F \times F}$ that is defined for pairs of histogram bins. The transport problem is a linear program with $z = f(x, \Theta)$ as in Equation 10. Please see Appendix Sections A.3.1 and A.3.3 for the assumptions made and the notations used for Algorithmic stability. We now derive single layer and SGD bounds.

### 4.0.1 SINGLE LAYER BOUND

By the triangle inequality,

$$|W_1(f(x, \Theta_S), y) - W_1(f(x, \Theta_{S'}), y)| \leq W_1(f(x, \Theta_S), f(x, \Theta_{S'})).$$

By (Villani, 2008, Theorem 6.13), recall that the diameter of the support is $D$, we have

$$W_1(f(x, \Theta_S), f(x, \Theta_{S'})) \leq D \|f(x, \Theta_S) - f(x, \Theta_{S'})\|_{\text{TV}}$$

where $\|\cdot\|_{\text{TV}}$ is the total variation distance, which by definition is

$$\|f(x, \Theta_S) - f(x, \Theta_{S'})\|_{\text{TV}} = \frac{1}{2}\sum_{i=1}^{F} |[f(x, \Theta_S)]_i - [f(x, \Theta_{S'})]_i| = \frac{1}{2}\|\sigma(E_x \cdot \Theta_S) - \sigma(E_x \cdot \Theta_{S'})\|_1$$

where $\|\cdot\|_1$ is the $L^1$-distance on $\mathbb{R}^F$. Since the softmax function is Lipschitz continuous, we have

$$\|f(x, \Theta_S) - f(x, \Theta_{S'})\|_{\text{TV}} \leq \frac{L_\sigma}{2}|E_x| \cdot \|\Theta_S - \Theta_{S'}\|_1$$

and thus

$$|\mathbb{E}_A[W_1(f(x,\Theta_S),y)-W_1(f(x,\Theta_{S'}),y)]|\leq \tfrac{L_\sigma D}{2}\sup_{x\in V}|E_x|\cdot\mathbb{E}_A\|\Theta_S-\Theta_{S'}\|_1=\tfrac{L_\sigma D}{2}g_\lambda\mathbb{E}_A\|\Theta_S-\Theta_{S'}\|_1 \quad (4)$$

where we used notation $g_\lambda := \sup_{x\in V}|E_x|$ as defined in Verma & Zhang (2019), which is known to be upper bounded by $\lambda_G^{\max}$, the spectrum of the graph Laplacian $L_G$ with largest absolute value.

### 4.0.2 SGD BOUND

It now remains to bound $\mathbb{E}_A\|\Theta_S-\Theta_{S'}\|_1$ resulting from the SGD iterations. The main technical challenge, as noted before, is to generalise the results to the Wasserstein space which is an abstract metric space without linear structure. Specifically, we have to modify the "gradient" in the Wasserstein space as the straightforward version Verma & Zhang (2019) does not satisfy the Lipschitz condition required in the algorithmic stability framework. To the best of our knowledge this modification is not seen in existing literature and can be thought of as a generalisation of the "gradient clipping" operation Hardt et al. (2016). The entire proof is in the Appendix A.3.4. We state the main result here.

$$\mathbb{E}_A\left[\|\Theta_{S,T}-\Theta_{S',T}\|_1\right]\leq \frac{2\eta F g_\lambda D}{m}\sum_{t=1}^{T}\left(1+\frac{3}{2}\eta D L_\sigma g_\lambda^2\right)^{t-1}. \quad (5)$$

Combining equation 4 and equation 23 gives us

$$|\mathbb{E}_A\left[W_1\left(f\left(x,\Theta_S\right),y\right)-W_1\left(f\left(x,\Theta_{S'}\right),y\right)\right]|\leq \frac{\eta F L_\sigma g_\lambda^2 D^2}{m}\sum_{t=1}^{T}\left(1+\frac{3}{2}\eta D L_\sigma g_\lambda^2\right)^{t-1}. \quad (6)$$

Therefore, we actually have the uniform algorithmic stability equation 14 holds with

$$\beta_m = \frac{\eta F L_\sigma g_\lambda^2 D^2}{2m}\sum_{t=1}^{T}\left(1+\frac{3}{2}\eta D L_\sigma g_\lambda^2\right)^{t-1}. \quad (7)$$

Note that here $\beta_m = O(\frac{1}{m})$ (needed to obtain a tight generalisation bound).

### 4.1 PUTTING EVERYTHING TOGETHER

**Lemma 1** *(Verma & Zhang, 2019, Lemma 4): Let $\lambda_G^{max}$ be the maximum absolute eigenvalue of $L_G$. Let $G_x$ be the ego-graph of a vertex $x\in V$ with corresponding maximum absolute eigenvalue $\lambda_{G_x}^{max}$. Then the following eigenvalue (singular value) bound holds $\forall x\in V$,*

$$\lambda_{G_x}^{max}\leq \lambda_G^{max} \quad (8)$$

**Lemma 2** *(Verma & Zhang, 2019, Theorem 2) A uniform stable randomised algorithm $(A_S,\beta_m)$ with a bounded loss function $0\leq \ell(A_S,\mathbf{y})\leq B$, satisfies the following generalisation bound with probability at least $1-\delta$, over the random draw of $S,\mathbf{y}$ with $\delta\in(0,1)$,*

$$\mathbb{E}_A\left[R(A_S)-R_{emp}(A_S)\right]\leq 2\beta_m + (4m\beta_m+B)\sqrt{\frac{\log\frac{1}{\delta}}{2m}} \quad (9)$$

where $R(A_S)$ is the generalisation error/risk and $R_{\text{emp}}(A_S)$ is the empirical error. Please see Appendix A.3.2 for definitions. Finally, by Equation 8, and Equation 9, our result i.e. Equation 7 immediately gives the following theorem:

**Theorem 1** *Let $A_S$ be a one-layer GNN algorithm (of 3) equipped with the graph convolutional filter $g(L_G)$ and trained on a dataset $S$ for $T$ iterations. Let the loss and activation functions be Lipschitz-continuous and smooth. Then the following expected generalisation gap holds with probability at least $1-\delta$, $\delta\in\{0,1\}$:*

$$\mathbb{E}_{SGD}\left[R(A_S)-R_{emp}(A_S)\right]\leq \frac{1}{m}O\left((\lambda_G^{max})^{2T}\right)+\left(O\left((\lambda_G^{max})^{2T}\right)+B\right)\sqrt{\frac{\log\frac{1}{\delta}}{2m}}$$

where the expectation $\mathbb{E}_{SGD}$ is taken over the randomness inherent in SGD, $m$ is the no. training samples, and $B$ is a constant which depends on the loss function. Our theorem states that GNN trained withe Wasserstein loss enjoys the same generalisation error bound of the traditional GCN (trained with cross entropy). We now discuss experiments.

## 5 EXPERIMENTS

Table 1: Statistics of datasets used in the experiments. Please see Section 3 for the notations used.

| Dataset | Type | $|\mathbf{V}|$ | $|\mathbf{E}|$ | $|\mathbf{E_d}|$ | F | Avg. edge size | Max. edge size |
|---|---|---|---|---|---|---|---|
| Cora | Co-authorship | 2653 | 2591 | 12071 | 7 | $2.3 \pm 1.9$ | 29 |
| DBLP | Co-authorship | 22535 | 43413 | 117215 | 5 | $4.7 \pm 6.1$ | 143 |
| Amazon | Recommendation | 84893 | 166994 | 1081994 | 5 | $3.0 \pm 3.0$ | 187 |
| ACM | Co-authorship | 67057 | 25511 | 59884 | 6 | $2.4 \pm 1.2$ | 32 |
| arXiv | Co-authorship | 790790 | 1354752 | 6728683 | 7 | $4.0 \pm 19.7$ | 2832 |

To demonstrate the effectiveness of our proposed DHN, we conducted experiments on 5 real-world directed hypergraphs. Four of them are Co-authorship datasets and one of them is a recommendation dataset. Table 1 shows the statistics of the datasets. For more details on the construction of the datasets, please see Appendix A.6.

### 5.1 EXPERIMENTAL SETUP

Inspired by the experimental setups of prior related works Kipf & Welling (2017); Liao et al. (2019), we tune hyperparameters using the Cora citation network dataset alone and use the optimal hyperparameters for all the other datasets. We hyperparameterise the cost matrix (base metric of the Wasserstein distance) as follows:

$$C = \begin{bmatrix} 1 & \eta & \eta & \dots & \eta & \eta \\ \eta & 1 & \eta & \dots & \eta & \eta \\ \vdots & \vdots & \vdots & \ddots & \vdots & \vdots \\ \eta & \eta & \eta & \dots & \eta & 1 \end{bmatrix}$$

The cost matrix $C$ is an $F \times F$ matrix ($F$ is the number of histogram bins) with ones on the diagonal and a hyperparameter $\eta$ elsewhere. We could have used a matrix of all $\eta$s. But it is no different from a matrix of all ones from the optimisation perspective and so we used the above more general matrix. Details of hyperparameter tuning and optimal hyperparameters are in Appendix A.5.

### 5.2 BASELINES

We used both Wasserstein distance and KL divergence to train different models. As already noted, we used the Sinkhorn algorithm to compute the (regularised) Wasserstein distance. Please see Appendix A.1 for more details. We compared DHN with the following baselines:

- **KL-MLP**: We used a simple multi-layer perceptron (MLP) on the features of the vertices and trained it using KL-divergence

- **OT-MLP**: We trained another MLP with the Wasserstein distance as the loss function. Note that this baseline and the previous baseline do not use the structure (graph / hypergraph)

- **KLR-MLP**: We regularised an MLP with explicit KL-divergence-based regularisation that uses the structure (graph / hypergraph) Subramanya & Bilmes (2011).

- **OTR-MLP**: We regularised an MLP with explicit Wasserestein-distance-based regularisation that uses the structure (graph / hypergraph) Solomon et al. (2014). For hypergraphs we used the clique expansion of the hypergraph Gao et al. (2019).

- **KL-HGNN / KL-HyperGCN**: We trained the different GCN-based methods on hypergraphs with KL divergence loss function on the labelled vertices.

- **Soft-HGNN / Soft-HyperGCN**: We trained the different GCN-based methods on hypergraphs with the Wasserstein distance as the loss function.

**Metric for comparison:** We use the mean squared error (MSE) between true and predicted distributions on the test set of vertices. Table 2 shows MSEs on the test split for all the three datasets.

Table 2: Results on real-world directed hypergraphs. We report $100\times$ mean squared errors (lower is better) over 10 different train-test splits. Note that all the reported numbers need to be multiplied by 0.01 to get the actual numbers. Please see section 5 for more details.

| Method | Cora | DBLP | ACM | Amazon Office Products | arXiv |
|---|---|---|---|---|---|
| KL-MLP | $8.94 \pm 0.16$ | $7.72 \pm 0.14$ | $8.47 \pm 0.15$ | $6.81 \pm 0.16$ | $10.87 \pm 0.25$ |
| OT-MLP | $7.45 \pm 0.35$ | $7.53 \pm 0.18$ | $7.85 \pm 0.26$ | $6.78 \pm 0.24$ | $10.01 \pm 0.23$ |
| KLR-MLP | $8.05 \pm 0.22$ | $7.35 \pm 0.18$ | $7.82 \pm 0.29$ | $6.74 \pm 0.15$ | $-$ |
| OTR-MLP | $6.57 \pm 0.43$ | $7.24 \pm 0.18$ | $6.77 \pm 0.32$ | $6.72 \pm 0.23$ | $-$ |
| KL-HGNN | $7.86 \pm 0.25$ | $7.17 \pm 0.12$ | $7.23 \pm 0.19$ | $6.71 \pm 0.19$ | $9.95 \pm 0.25$ |
| KL-HyperGCN | $7.95 \pm 0.27$ | $7.15 \pm 0.17$ | $7.53 \pm 0.21$ | $6.69 \pm 0.17$ | $9.99 \pm 0.23$ |
| Soft-HGNN | $5.97 \pm 0.37$ | $6.18 \pm 0.37$ | $6.02 \pm 0.37$ | $6.63 \pm 0.39$ | $8.61 \pm 0.49$ |
| Soft-HyperGCN | $6.02 \pm 0.32$ | $6.21 \pm 0.35$ | $6.04 \pm 0.32$ | $6.61 \pm 0.30$ | $8.60 \pm 0.47$ |
| KL-DHN (ours) | $7.04 \pm 0.24$ | $6.97 \pm 0.22$ | $7.16 \pm 0.24$ | $6.65 \pm 0.17$ | $9.34 \pm 0.32$ |
| Soft-DHN (ours) | $\mathbf{4.87 \pm 0.40}$ | $\mathbf{5.65 \pm 0.42}$ | $\mathbf{5.12 \pm 0.34}$ | $\mathbf{6.55 \pm 0.33}$ | $\mathbf{7.69 \pm 0.36}$ |

## 5.3 DISCUSSION

We used a simple one-layer architecture for our proposed DHN and a 2-hop simplified GCN Wu et al. (2019a) as the GNN model on the graph $\mathcal{G} = (E, E_d)$ i.e.

$$Z = softmax(I \cdot H_E \cdot \Theta_1), \quad H_E = A^2 X_E \Theta_2$$

where $A$ is the symmetically normalised adjacency (with self loops) of the graph $\mathcal{G}$. We demonstrate that this simple model is effective enough through an ablation study in Appendix Table 4. Our results demonstrate strong performances across all the datasets esp. on the co-authorships networks.

Specifically, we observe that Soft models (that use the Wasserstein loss) are almost always superior to their counterparts that use the KL divergence as the loss function. This is because the Soft models can exploit the distance matrix $C$ while KL-divergence does not.

Moreover, our proposed DHN outperforms two strong hypergraph baselines viz. HGNN Feng et al. (2019) and HyperGCN Yadati et al. (2019). We believe this is because they do not exploit the rich structural information in the directed hyperedges (connections among hyperedges) while our proposed DHN does exploit them.

We also experimented on standard graph benchmark node-classification datasets such as Cora, Citeseer, and Pubmed by treating the class label as one-hot probability distribution. We used the Soft variants of GCN Kipf & Welling (2017), Simple GCN Wu et al. (2019a), and GAT Veličković et al. (2018). We achieved competitive results as shown in Appendix Table 3

## 6 CONCLUSION

We have proposed DHN, a novel method for soft SSL on directed hypergraphs. DHN can effectively propagate histograms to unknown vertices by integrating vertex features, directed hyperedges and undirected hypergraph structure. As a key contribution, we have established generalisation bounds for DHN within the framework of algorithmic stability. We have also demonstrated DHN's effectiveness through detailed experimentation on real-world hypergraph datasets.

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

## A   Appendix

The appendix is organised as follows:

1. Regularised Wasserstein and the Sinkhorn algorithm
2. Time complexity of the proposed DHN
3. Proofs and more notations used for theoretical analysis
4. Additional experiments (Graph-based soft SSL and Ablation Study)
5. Details of hyperparameters
6. Sources of real-world datasets

### A.1   Regularised Wasserstein and the Sinkhorn Algorithm

For discrete distributions, $W_p$ of Equation 2 is the solution of a linear program:

$$W_p\left(z,y\right)^p = \min \sum_{i=1}^{F} \sum_{j=1}^{F} C_{ij}^p \pi_{ij}$$

$$\text{s.t.} \quad \pi_{ij} \geq 0, \sum_{j=1}^{F} \pi_{ij} = [z]_i, \sum_{i=1}^{F} \pi_{ij} = [y]_j \qquad \forall 1 \leq i,j \leq F \tag{10}$$

where for an arbitrary element $q \in \mathcal{P}_F$, $[q]_i$ stands for the probability mass in the $i$th bin.

#### A.1.1   Sinkhorn divergence

The expensive linear program 10 can be efficiently solved by entropic regularisation Cuturi (2013):

$$W_p^\lambda(z,y)^p = \min \sum_{i=1}^{F} \sum_{j=1}^{F} C_{ij}^p \pi_{ij} + \lambda \cdot \text{Tr}\left( \pi \left( \log \pi - \mathbf{1}\mathbf{1}^T \right)^T \right) \tag{11}$$

$$\text{s.t.} \quad \pi_{ij} \geq 0, \sum_{j=1}^{F} \pi_{ij} = [z]_i, \sum_{i=1}^{F} \pi_{ij} = [y]_j \qquad \forall 1 \leq i,j \leq F$$

where $\log(\cdot)$ is applied element-wise and $\lambda \geq 0$ is a hyperparameter. The optimal solution $\pi^*$ for $\lambda > 0$ takes the following form.

$$\pi^* = \text{diag}(r) \cdot \exp\left( -\frac{C^p}{\lambda} \right) \cdot \text{diag}(c)$$

where $\text{diag}(\mathbf{z})$ is a diagonal matrix with the components of $\mathbf{z}$ in the diagonal places.

**Sinkhorn algorithm Cuturi (2013):** We optimise Equation 11 for $r$ and $c$ via matrix balancing, i.e., start with an initial $K := \exp(-\frac{C^p}{\lambda})$ and alternately ensure the marginal constraints are satisfied until convergence:

$$r \leftarrow z./(Kc) \qquad c \leftarrow y./(K^T r)$$

where $./$ is element-wise division. We use the above efficient algorithm in our experiments.

## A.2 TIME COMPLEXITY OF DHN

We consider the problem of predicting probability distributions for the vertices in $\mathcal{H} = (V, E_d)$ given a typically small subset $V_k \subseteq V$ of vertices with known distributions. In this work, we are concerned with discrete distributions modelled on a complete separable metric space $(M, C)$. Furthermore, we assume that we are provided with a feature matrix, $X_V \in \mathbb{R}^{n \times D_V}$, in which each vertex $v \in V$ is represented by a $D_v$-dimensional feature vector $x_v$ (here $n = |V|$). We are also provided with a hyperedge feature matrix $X_E \in \mathbb{R}^{m \times D_E}$ with $x_e, e \in E$ as $D_E$-dimensional feature representations. Let $T$ be the time taken for the Sinkhorn algorithm on all the vertices with known distributions. Note that $T$ can be approximated in near-linear time Altschuler et al. (2017). Further, let $T$ be the total number of epochs of training. Define

$$N := \sum_{e \in E} |e|, \qquad N_c := \sum_{e \in E} {}^{|e|}C_2$$

The time complexity oa one-layer DHN is $O\Big(|E_d| \cdot D_E \cdot h1 + |E| \cdot D_V \cdot h_2\Big)$ where $h_1$ is the number of hidden units of the GNN layer and $h_2$ is the number of output channels.

## A.3 THEORETICAL ANALYSIS: PROOFS

This section is organised as follows:

1. Assumptions/notations
2. Definitions of generalisation and empirical errors
3. Framework of Algorithmic Stability
4. Proof of SGD Bound

### A.3.1 ASSUMPTIONS/NOTATIONS

To avoid unnecessary technical complications, assume the histogram admits a geometric realisation over the one-dimensional Euclidean space, such that the $i$th bin is placed at location $b_i \in \mathbb{R}$, and set

$$C_{ij} := |b_i - b_j|, \quad \forall 1 \le i, j \le F.$$

Without loss of generality, we assume $b_1 \le b_2 \le \cdots \le b_F$, and write $h_i := b_{i+1} - b_i \ge 0$ for all $i = 1, \ldots, F - 1$. Denote the diameter of the support by $D := \max_{1 \le i, j \le F} |b_i - b_j| = b_F - b_1$. We take the Wasserstein cost as the Wasserstein-1 distance: $W(\mu, \nu) := W_1(\mu, \nu)$. In this particular one-dimensional setting, we have a particularly simple form for the cost function:

$$W_1(\mu, \nu) = \int_0^1 |F_\mu^{-1}(s) - F_\nu^{-1}(s)| \, \mathrm{d}s = \int_{-\infty}^\infty |F_\mu(t) - F_\nu(t)| \, \mathrm{d}t \tag{12}$$

where $F_\mu : \mathbb{R} \to [0, 1]$, $F_\nu : \mathbb{R} \to [0, 1]$ are the cumulative distribution functions of $\mu$, $\nu$, respectively; $F_\mu^{-1}$, $F_\nu^{-1}$ are the *generalized inverses* of $F_\mu$ and $F_\nu$, respectively, defined as (similar for $F_\nu^{-1}$)

$$F_\mu^{-1}(t) := \inf \{b \in \mathbb{R} : F_\mu(b) > t\}, \qquad \forall t \in [0, 1]. \tag{13}$$

This characterisation is seen in any standard literature on optimal transport, e.g., (Villani, 2003, Theorem 2.18).

### A.3.2 DEFINITIONS: GENERALISATION AND EMPIRICAL ERRROS

Let the learning algorithm $A_S$ on a dataset $S$ be a function from $\zeta^m$ to $(\mathcal{Y})^X$. where $\mathcal{X}$ is the input Hilber space, $\mathcal{Y}$ is the output Hilbert space, $\zeta = \mathcal{X} \times \mathcal{Y}$. The training set of datapoints, labels is $S = \{z_1 = (x_1, y_1), \cdots, z_k = (x_k, y_k))\}$. Let the loss function be $\ell : \zeta^m \times \zeta \to \mathbb{R}$. Then the generalisation error or risk $R(A_S)$ is deined as

$$R(A_S) := \mathbb{E}\Big[\ell(A_S, z)\Big] = \int \ell(A_S, z) p(z) dz$$

where $p(z)$ is the probability of seeing the sample $z \in S$.

The empirical error, on the other hand, is defined as

$$R_{\text{emp}}(A_S) := \frac{1}{k} \sum_{j=1}^{k} \ell(A_S, z_j)$$

### A.3.3 ALGORITHMIC STABILITY

Denote $S \subset V \times \mathcal{P}_F$ for an arbitrary training data set sampled with respect to distribution $\mathcal{D}$. A learning algorithm for GNN, denoted as $A$, maps a training set $S$ to a trained GNN $f(\cdot, \Theta_S) : x \mapsto f(x, \Theta_S)$. Let $S'$ be another training data set that differs from $S$ by exactly one data. Our goal in this section is to establish a uniform stability

$$\sup_{\substack{S \subset V \times \mathcal{P}_F \\ (x,y) \in V \times \mathcal{P}_F}} |\mathbb{E}_A \left[ \ell \left( f(\cdot, \Theta_S), (x, y) \right) \right] - \mathbb{E}_A \left[ \ell \left( f(\cdot, \Theta_{S'}), (x, y) \right) \right]|$$

$$= \sup_{\substack{S \subset V \times \mathcal{P}_F \\ (x,y) \in V \times \mathcal{P}_F}} |\mathbb{E}_A \left[ W_1 \left( f(x, \Theta_S), y \right) \right] - \mathbb{E}_A \left[ W_1 \left( f(x, \Theta_{S'}), y \right) \right]|$$

$$= \sup_{\substack{S \subset V \times \mathcal{P}_F \\ (x,y) \in V \times \mathcal{P}_F}} |\mathbb{E}_A \left[ W_1 \left( f(x, \Theta_S), y \right) - W_1 \left( f(x, \Theta_{S'}), y \right) \right]| \leq 2\beta_m \tag{14}$$

with $\beta_m = O(1/m)$, which in turn can be used to establish the generalisation error bound of the form (Verma & Zhang, 2019, Theorem 1), following the framework Hardt et al. (2016) for SGD-based learning algorithms. Following the arguments in Verma & Zhang (2019), this boils down to checking a few Lipschitz properties for the cost function.

### A.3.4 SGD BOUND: PROOF

Note that

$$W_1(\mu, \nu) = \int_0^1 \left| F_\mu^{-1}(s) - F_\nu^{-1}(s) \right| ds = \int_{-\infty}^{\infty} |F_\mu(t) - F_\nu(t)| dt \tag{15}$$

and from the single-layer bound

$$|\mathbb{E}_A[W_1(f(x, \Theta_S), y) - W_1(f(x, \Theta_{S'}), y)]| \leq \frac{L_\sigma D}{2} \sup_{x \in V} |E_x| \cdot \mathbb{E}_A \|\Theta_S - \Theta_{S'}\|_1 = \frac{L_\sigma D}{2} g_\lambda \mathbb{E}_A \|\Theta_S - \Theta_{S'}\|_1 \tag{16}$$

Uniform stability:

$$\sup_{\substack{S \subset V \times \mathcal{P}_F \\ (x,y) \in V \times \mathcal{P}_F}} |\mathbb{E}_A \left[ \ell \left( f(\cdot, \Theta_S), (x, y) \right) \right] - \mathbb{E}_A \left[ \ell \left( f(\cdot, \Theta_{S'}), (x, y) \right) \right]|$$

$$= \sup_{\substack{S \subset V \times \mathcal{P}_F \\ (x,y) \in V \times \mathcal{P}_F}} |\mathbb{E}_A \left[ W_1 \left( f(x, \Theta_S), y \right) \right] - \mathbb{E}_A \left[ W_1 \left( f(x, \Theta_{S'}), y \right) \right]|$$

$$= \sup_{\substack{S \subset V \times \mathcal{P}_F \\ (x,y) \in V \times \mathcal{P}_F}} |\mathbb{E}_A \left[ W_1 \left( f(x, \Theta_S), y \right) - W_1 \left( f(x, \Theta_{S'}), y \right) \right]| \leq 2\beta_m \tag{17}$$

It now remains to bound $\mathbb{E}_A \|\Theta_S - \Theta_{S'}\|_1$ resulting from the SGD iterations. Given training set $S$, applying SGD to GNN amounts to performing the updates

$$\Theta_{S,t+1} = \Theta_{S,t} - \eta \nabla_\Theta \ell(f(\cdot, \Theta), (x_{i_t}, y_{i_t})) = \Theta_{S,t} - \eta \nabla_\Theta W_1(f(x_{i_t}, \Theta_{S,t}), y_{i_t})$$

where $\eta > 0$ is the learning rate and $z_{i_t} = (x_{i_t}, y_{i_t})$ are random data i.i.d. uniformly sampled from the training set. By the simple formulae equation 15 for one-dimensional optimal transport, we can explicitly write out for any parameter set $\Theta$ and data $z = (x, y)$

$$
W_1\left(f\left(x, \Theta\right), y\right) = \int_{-\infty}^{\infty} \left| F_{f(x,\Theta)}\left(t\right) - F_y\left(t\right) \right| \mathrm{d}t = \sum_{i=1}^{F-1} \left(x_{i+1} - x_i\right) \left| \sum_{j=1}^{i} \left(\left[f\left(x, \Theta\right)\right]_i - \left[y\right]_i\right) \right|
$$

$$
= \sum_{i=1}^{F-1} h_i \left| \sum_{j=1}^{i} \left(\left[\sigma\left(E_x \cdot \Theta\right)\right]_i - \left[y\right]_i\right) \right|
$$

where again we used notation $[y]_i$ to denote the probability mass of $y \in \mathcal{P}_F$ in the $i$th bin, for all $i = 1, \dots, F$. Thus

$$
\frac{\partial}{\partial \Theta_k} W_1\left(f\left(x, \Theta\right), y\right) = \sum_{i=1}^{F-1} h_i \cdot \mathrm{sgn} \left\{ \sum_{j=1}^{i} \left(\left[\sigma\left(E_x \cdot \Theta\right)\right]_j - \left[y\right]_j\right) \right\} \cdot \sum_{j=1}^{i} \frac{\partial}{\partial \Theta_k} \left[\sigma\left(E_x \cdot \Theta\right)\right]_j
$$

$$
= \sum_{i=1}^{F-1} h_i \cdot \mathrm{sgn} \left\{ \sum_{j=1}^{i} \left(\left[\sigma\left(E_x \cdot \Theta\right)\right]_j - \left[y\right]_j\right) \right\} \cdot E_x \sum_{j=1}^{i} \left[\sigma\left(E_x \cdot \Theta\right)\right]_j \left(\delta_{jk} - \left[\sigma\left(E_x \cdot \Theta\right)\right]_k\right)
$$

$$
= E_x \cdot \left[\sigma\left(E_x \cdot \Theta\right)\right]_k \sum_{i=1}^{F-1} h_i \cdot \mathrm{sgn} \left\{ \sum_{j=1}^{i} \left(\left[\sigma\left(E_x \cdot \Theta\right)\right]_j - \left[y\right]_j\right) \right\} \left(1 - \sum_{j=i}^{j} \left[\sigma\left(E_x \cdot \Theta\right)\right]_j\right)
$$

$$
= E_x \cdot \left[\sigma(E_x \cdot \Theta)\right]_k \sum_{i=1}^{F-1} h_i \cdot \mathrm{sgn}\left\{\sum_{j=1}^{i}\left(\left[\sigma(E_x \cdot \Theta)\right]_j - [y]_j\right)\right\}\left(\sum_{j=i}^{j}\left([y]_j - \left[\sigma(E_x \cdot \Theta)\right]_j\right) + \sum_{j=i+1}^{F}[y]_j\right)
$$

$$
= -E_x \cdot \left[\sigma(E_x \cdot \Theta)\right]_k \cdot W_1(\sigma(E_x \cdot \Theta), y) + E_x \cdot \left[\sigma(E_x \cdot \Theta)\right]_k \sum_{i=1}^{F-1} h_i \cdot \mathrm{sgn}\left\{\sum_{j=1}^{i}\left(\left[\sigma(E_x \cdot \Theta)\right]_j - [y]_j\right)\right\} \sum_{j=i+1}^{F}[y]_j
$$

where $\mathrm{sgn}\left\{\cdot\right\}$ is the sign function, and $\delta_{ik}$ is the Kronecker delta notation. The second equality used the specific form of the derivative of the softmax function. Unfortunately, this gradient is not Lipschitz continuous due to the sign function in the second term. Nevertheless, if we use a "modified gradient" that drops the second term, i.e., choose to update the parameter $\Theta_k$ in the direction

$$
\widetilde{\frac{\partial}{\partial \Theta_k}} W_1\left(f\left(x, \Theta\right), y\right) := -E_x \cdot \left[\sigma\left(E_x \cdot \Theta\right)\right]_k \cdot W_1\left(\sigma\left(E_x \cdot \Theta\right), y\right) \tag{18}
$$

then obviously this new choice of "descent" direction is certainly Lipschitz continuous, as

$$
\left| \widetilde{\frac{\partial}{\partial \Theta_k}} W_1\left(f\left(x, \Theta_S\right), y\right) - \widetilde{\frac{\partial}{\partial \Theta_k}} W_1\left(f\left(x, \Theta_{S'}\right), y\right) \right|
$$

$$
= |E_x| \cdot \left| \left[\sigma\left(E_x \cdot \Theta_S\right)\right]_k \cdot W_1\left(\sigma\left(E_x \cdot \Theta_S\right), y\right) - \left[\sigma\left(E_x \cdot \Theta_{S'}\right)\right]_k \cdot W_1\left(\sigma\left(E_x \cdot \Theta_{S'}\right), y\right) \right|
$$

$$
\leq |E_x| \cdot \left[ \left| \left[\sigma\left(E_x \cdot \Theta_S\right)\right]_k - \left[\sigma\left(E_x \cdot \Theta_{S'}\right)\right]_k \right| \cdot W_1\left(\sigma\left(E_x \cdot \Theta_S\right), y\right) \right.
$$

$$
\left. + \left[\sigma\left(E_x \cdot \Theta_{S'}\right)\right]_k \cdot \left| W_1\left(\sigma\left(E_x \cdot \Theta_S\right), y\right) - W_1\left(\sigma\left(E_x \cdot \Theta_{S'}\right), y\right) \right| \right]
$$

$$
\leq |E_x| \cdot \left( L_\sigma D |E_x| \cdot \|\Theta_S - \Theta_{S'}\|_1 + W_1\left(\sigma\left(E_x \cdot \Theta_S\right), \sigma\left(E_x \cdot \Theta_{S'}\right)\right) \right)
$$

$$
\leq |E_x| \cdot \frac{3}{2} L_\sigma D \cdot |E_x| \cdot \|\Theta_S - \Theta_{S'}\|_1 \leq \frac{3}{2} L_\sigma D \cdot g_\lambda^2 \|\Theta_S - \Theta_{S'}\|_1.
$$

Therefore, if we define

$$
\widetilde{\nabla}_\Theta W_1\left(f\left(x, \Theta\right), y\right) := \left( \widetilde{\frac{\partial}{\partial \Theta_1}} W_1\left(f\left(x, \Theta\right), y\right), \cdots, \widetilde{\frac{\partial}{\partial \Theta_F}} W_1\left(f\left(x, \Theta\right), y\right) \right)^\top \tag{19}
$$

then the stochastic update algorithm[1]

$$
\Theta_{S,t+1} = \Theta_{S,t} - \eta \widetilde{\nabla}_\Theta W_1\left(f\left(x_{i_t}, \Theta_{S,t}\right), y_{i_t}\right) \tag{20}
$$

---

[1]Note that this is not even a stochastic gradient descent algorithm! the "gradient" involved is a "fake" gradient — this is the counter-intuitive part.

will still satisfy the generalization bound, due to the Lipschitz continuity

$$\left\| \widetilde{\nabla}_{\Theta} W_1\left(f\left(x, \Theta_S\right), y\right) - \widetilde{\nabla}_{\Theta} W_1\left(f\left(x, \Theta_{S'}\right), y\right) \right\|_1 \leq \frac{3D}{2} L_\sigma g_\lambda^2 \left\| \Theta_S - \Theta_{S'} \right\|_1. \tag{21}$$

In fact, equation 21 is exactly the GNN analogy of (Verma & Zhang, 2019, Lemma 1) which establishes the stability for the "same sample loss" case. The "different sample loss" analogy, or the lemma in (Verma & Zhang, 2019, Lemma 2), can be trivially obtained by the definition equation 18. In fact, noting that

$$\left| \frac{\widetilde{\partial}}{\partial \Theta_k} W_1\left(f\left(x, \Theta\right), y\right) \right| = \left| E_x \cdot \left[\sigma\left(E_x \cdot \Theta\right)\right]_k \cdot W_1\left(\sigma\left(E_x \cdot \Theta\right), y\right) \right| \leq g_\lambda D,$$

we easily obtain

$$\left| \frac{\widetilde{\partial}}{\partial \Theta_k} W_1\left(f\left(x, \Theta_S\right), y\right) - \frac{\widetilde{\partial}}{\partial \Theta_k} W_1\left(f\left(x', \Theta_{S'}\right), y'\right) \right| \leq 2g_\lambda D$$

and it follows that

$$\left\| \widetilde{\nabla}_{\Theta} W_1\left(f\left(x, \Theta_S\right), y\right) - \widetilde{\nabla}_{\Theta} W_1\left(f\left(x, \Theta_{S'}\right), y\right) \right\|_1 \leq 2F g_\lambda D. \tag{22}$$

Putting together equation 21 and equation 22, we obtain the following analogy of (Verma & Zhang, 2019, Lemma 3): Starting with two training data sets $S$, $S'$ that differs by exactly one sample, after each iteration $t$ we have

$$\mathbb{E}_A\left[ \left\| \Theta_{S,t+1} - \Theta_{S',t+1} \right\|_1 \right] = \mathbb{E}_A\left[ \left\| \Theta_{S,t} - \eta \widetilde{\nabla}_{\Theta} W_1\left(f\left(x_t, \Theta_{S,t}\right), y_t\right) - \Theta_{S',t} + \eta \widetilde{\nabla}_{\Theta} W_1\left(f\left(x'_t, \Theta_{S',t}\right), y'_t\right) \right\| \right]$$

$$\leq \mathbb{E}_A\left[ \left\| \Theta_{S,t+1} - \Theta_{S',t+1} \right\|_1 \right] + \left(1 - \tfrac{1}{m}\right) \cdot \eta \cdot \tfrac{3D}{2} L_\sigma g_\lambda^2 \mathbb{E}_A\left[ \left\| \Theta_{S,t} - \Theta_{S',t} \right\|_1 \right] + \tfrac{1}{m} \cdot \eta \cdot 2F g_\lambda D$$

$$\leq \left(1 + \frac{3}{2}\eta D L_\sigma g_\lambda^2\right) \mathbb{E}_A\left[ \left\| \Theta_{S,t} - \Theta_{S',t} \right\|_1 \right] + \frac{2\eta F g_\lambda D}{m}.$$

Solving this first-order recursion gives the stability after $T$ random update steps:

$$\mathbb{E}_A\left[ \left\| \Theta_{S,T} - \Theta_{S',T} \right\|_1 \right] \leq \frac{2\eta F g_\lambda D}{m} \sum_{t=1}^{T} \left(1 + \frac{3}{2}\eta D L_\sigma g_\lambda^2\right)^{t-1}. \tag{23}$$

Combining equation 16 and equation 23 gives us

$$\left| \mathbb{E}_A\left[ W_1\left(f\left(x, \Theta_S\right), y\right) - W_1\left(f\left(x, \Theta_{S'}\right), y\right) \right] \right| \leq \frac{\eta F L_\sigma g_\lambda^2 D^2}{m} \sum_{t=1}^{T} \left(1 + \frac{3}{2}\eta D L_\sigma g_\lambda^2\right)^{t-1}. \tag{24}$$

Therefore, we actually have the uniform algorithmic stability equation 17 holds with

$$\beta_m = \frac{\eta F L_\sigma g_\lambda^2 D^2}{2m} \sum_{t=1}^{T} \left(1 + \frac{3}{2}\eta D L_\sigma g_\lambda^2\right)^{t-1}. \tag{25}$$

## A.4 Additional Experiments

| Dataset | GCN | Soft-GCN | GAT | Soft-GAT | Simple-GCN | Soft-Simple-GCN |
|---|---|---|---|---|---|---|
| **Cora** | 81.5 | 81.7 | 82.8 | 82.5 | 81 | 81.6 |
| **Citeseer** | 70.3 | 70.1 | 70.6 | 70.6 | 71.8 | 71.6 |
| **Pubmed** | 79 | 78.6 | 78.7 | 78.4 | 78.8 | 78.8 |

Table 3: Accuracy on traditional graph-based SSL datasets. The experimental setting is the same as in GCN Kipf & Welling (2017) and GAT Veličković et al. (2018). We used $10\%$ of the labelled vertices as validation data to tune the hyperparameter $\eta$.

### A.4.1 Ablation Study

In this section, we compared our proposed one-layer DHN against deeper layers and without exploiting directed hyperedges.

Table 4: Ablation study of our proposed Soft-DHN. Please see section 5 for more details.

| # DHN layers | # GNN layers(hops) | Cora | DBLP |
|:---:|:---:|:---:|:---:|
| 2 | 2 | $7.68 \pm 0.24$ | $7.98 \pm 0.27$ |
| 2 | 1 | $7.64 \pm 0.25$ | $7.93 \pm 0.28$ |
| 2 | 0 | $7.69 \pm 0.27$ | $7.98 \pm 0.22$ |
| 1 | 0 | $5.64 \pm 0.32$ | $6.45 \pm 0.38$ |
| 1 | 1 | $5.41 \pm 0.35$ | $6.26 \pm 0.32$ |
| 1 | 2 | $\mathbf{4.87 \pm 0.40}$ | $\mathbf{5.65 \pm 0.42}$ |

## A.5 DETAILS OF HYPERPARAMETERS

Inspired by the experimental setups of prior related works Kipf & Welling (2017); Liao et al. (2019), we tune hyperparameters using the Cora co-authorship network dataset alone. The optimal hyperparameters are fixed and then used for all the other datasets. Table 5 shows the list of hyperparameters used in the datasets. Prior works Kipf & Welling (2017); Liao et al. (2019) have extensively performed tuning of hyperparameters such as hidden size, learning rate, etc and we fixed their reported optimal hyperparameters. It should be noted that self training and co-training methods Li et al. (2018b) can also be used in case of absence of validation data. We hyperparameterise the cost matrix (base metric of the Wasserstein distance) as follows:

$$C = \begin{bmatrix} 1 & \eta & \eta & \dots & \eta & \eta \\ \eta & 1 & \eta & \dots & \eta & \eta \\ \vdots & \vdots & \vdots & \ddots & \vdots & \vdots \\ \eta & \eta & \eta & \dots & \eta & 1 \end{bmatrix}$$

Table 6 shows the best results on the validation split of Cora (with optimal hyperaparameters). The training set had $140$ vertices, the validation set $1000$ vertices and the rest of the vertices were used to test the models. The results reported are after $200$ epochs of training with a seed value of $598$.

Table 5: List of hyperparameters used in the experiments. A set of values indicates that the corresponding hyperparameter is tuned from the set (on the validation split).

| Hyperparameter | Value(s) | Hyperparameter | Value(s) |
|:---|:---|:---|:---|
| hidden size | 16 | $\epsilon$ (Sinkhorn) | 0.1 |
| learning rate | 0.01 | # Sinkhorn iterations | 100 |
| dropout | 0.5 | $\eta$ | $\{0, 1, 2, \cdots, 40\}$ |
| weight decay | $5 \times 10^{-4}$ | $\lambda$ | $\{1, 0.5, 0.1, 0.05, 0.01, \cdots, 5 \times 10^{-7}, 10^{-7}\}$ |

## A.6 SOURCES OF THE REAL-WORLD DATASETS

**Co-authorship data**: All authors co-authoring a paper are in one hyperedge. We used the author data[2] to get the co-authorship hypergraph for cora. We manually constructed the DBLP dataset from Arnetminer[3].

---

[2]https://people.cs.umass.edu/ mccallum/data.html

[3]https://aminer.org/lab-datasets/citation/DBLP-citation-Jan8.tar.bz

Table 6: Optimal hyperparameters on the validation set on Cora Co-authorship network.

| Method | Optimal hyperparameters | Best MSE on validation set |
|---|---|---|
| KL-MLP | - | 7.87 |
| OT-MLP | $\eta = 31$ | 6.47 |
| KLR-MLP | - | 7.39 |
| OTR-MLP | $\eta = 25, \lambda = 5 \times 10^{-3}$ | 4.86 |
| KL-HGNN | - | 6.98 |
| KL-HyperGCN | - | 7.03 |
| Soft-HGNN | $\eta = 20$ | 3.24 |
| Soft-HyperGCN | $\eta = 17$ | 4.02 |
| KL-DHN | - | 6.34 |
| Soft-DHN | $\eta = 19$ | **2.67** |

### A.6.1 CONSTRUCTION OF THE DBLP DATASET

We downloaded the entire dblp data from `https://aminer.org/lab-datasets/citation/DBLP-citation-Jan8.tar.bz` Tang et al. (2008). The steps for constructing the dblp dataset used in the paper are as follows:

- We defined a set of 5 conference categories (histograms for the SSL task) as "algorithms", "database", "datamining", "intelligence", and "vision"
- For a total of 4304 venues in the entire dblp dataset we took papers from only a subset of venues from `https://en.wikipedia.org/wiki/List_of_computer_science_conferences` corresponding to the above 5 conferences
- From the venues of the above 5 conference categories, we got 22535 authors publishing at least two documents for a total of 43413
- We took the abstracts of all these 43413 documents, constructed a dictionary of the most frequent words (words with frequency more than 100) and this gave us a dictionary size of 1425
- We then extracted the 117215 citation links among these documents

### A.6.2 CONSTRUCTION OF THE AMAZON OFFICE PRODUCT DATASET

We downloaded the entire Amazon data He & McAuley (2016); McAuley et al. (2015). The steps for constructing the dataset used in the paper are as follows:

- We downloaded the office product ratings subset from the entire dataset
- We constructed a hypergraph of items with each hyperedge representing a user connecting all the items that they bought
- We removed hyperedges of size 1
- We connected a pair of hyperedges (bi-directional) if they had more than 1 item in common

### A.6.3 CONSTRUCTION OF THE ACM DATASET

We downloaded the entire ACM data from `https://lfs.aminer.org/lab-datasets/citation/acm.v9.zip` Tang et al. (2008). The steps for curating the dataset in the paper are as follows:

- Based on the number of papers published, we identified the six most popular venues: "Journal of Computational Physics", "IEEE Transactions on Pattern Analysis and Machine Intelligence", "Automatica (Journal of IFAC)", "IEEE Transactions on Information Theory", "Expert Systems with Apllications: An International Journal", and "IEEE Transactions on Computers"

- We then listed the set of all authors published in these venues (we got a total of $67057$ authors).

- We finally obtained the citation relationships of all the documents co-authored by these authors (total number of documents is $25511$ and total number of citations is $59884$)

### A.6.4 DETAILS OF THE ARXIV DATASET

We downloaded the entire arXiv dataset Clement et al. (2019) from `https://github.com/mattbierbaum/arxiv-public-datasets/releases/tag/v0.2.0`. The steps for curating the dataset in the paper are as follows:

- We removed papers without any authors and got a total of $13,54,752$ edges

- We extracted $67,28,683$ citation edges among these papers

- The total number of authors in these papers is $7,90,790$

