# OpenReview forum: "Graph Neural Networks for Soft Semi-Supervised Learning on Hypergraphs"
_ICLR.cc/2020/Conference — Reject_

### Official Review · AnonReviewer3 · 2019-10-23
**Official Blind Review #3**

**Rating:** 3

**Review:**

This work explores hypergraph-based SSL of histograms. DHN enables soft SSL learning based on the existing tools from optimal transportation. The idea of treating hyperedges as vertexes of another graph is novel and the theoretical analysis is sound.

However, the paper has the following issues:

1) The writing is poor, especially about the reference, and the description of how and why DHN works (section 3.3).

2) The novelty is restricted. It seems that the only innovation is introducing the information from the hyperedges into $H_E^{(0)}$.

3) Though the experiments on Cora and DBLP have revealed the superiority of DHN, the authors still need a more thorough empirical evaluation on some challenging benchmarks to draw the conclusion.

I'm willing to increase my score if the concerns are addressed.


**Experience Assessment:**

I have read many papers in this area.

**Review Assessment: Checking Correctness Of Derivations And Theory:**

I carefully checked the derivations and theory.

**Review Assessment: Checking Correctness Of Experiments:**

I carefully checked the experiments.

**Review Assessment: Thoroughness In Paper Reading:**

I read the paper thoroughly.

---

> ### Author Response · Authors · 2019-11-15
> **Our response to Reviewer #3**
>
> Thanks for the review.
>
> $\textbf{On an improved writing}:$
> Thanks for pointing this out. Reviewer #1 had similar concerns regarding paper writing in section 3.3 and references. We have improved the writing in the revised version.
>
>
>
> $\textbf{On the novelty of the paper:}$
> To the best of our knowledge, we are the first to explore GNNs for soft semi-supervised learning. We propose a novel method for directed hypergraphs. On the empirical side, we provide five benchmark datasets for soft semi-supervised learning on directed hypergraphs.
>
> On the theoretical front, the main novelty is to provide bounds for learning problem “valued in the Wasserstein space”. This requires some technicality as the Wasserstein space is an abstract metric space without linear structure.
>
> Specifically, we have to modify the “gradient” in the Wasserstein space as the straightforward version does not satisfy the Lipschitz condition required in the algorithmic stability framework. This modification is not seen in existing literature to the best of our knowledge. It can be thought of as a generalisation of the “gradient clipping” operation in [Hardt et al., ICML'16].
>
>
>
> $\textbf{On more benchmark datasets:}$
> Following the reviewer’s suggestion, we experimented on two additional benchmark datasets (ACM and arXiv) to evaluate the proposed method. The results are shown in Table 2 of our revised paper. As we can see, our method is consistently superior to baselines.
>
>
>
> [Hardt et al., ICML'16] Train faster, generalize better: Stability of stochastic gradient descent

---

### Official Review · AnonReviewer1 · 2019-10-25
**Official Blind Review #1**

**Rating:** 3

**Review:**

Contributions:
1. This paper proposes a semi-supervised graph neural network method for hypergraphs.
2. A generalization error bound was proposed adapted to the semi-supervised setting with Wasserstein's loss.
3. Empirical results demonstrate the effectiveness of the proposed method.

The algorithmic contribution of this paper is clear: it proposes a new network architecture that (1) initialize latent features H^{(0)}_E for hyperedges from a "hyperedge graph" and (2) learn latent features for each node and hyperedge using a GCN type network. Since the latent features are formulated as discrete distributions, a Wasserstein distance can be applied for training with the sinkhorn approximation algorithm.

I think the weakness of this paper is two folds, which makes it not ready to publish. First, the paper claims that the performance gain results from the exploitation of directed hyperedges. This is reasonable if I barely see the results for Soft-DHN. However, I find the design of the hyperedge GNN to be fragile regarding the number of layers from the result in Table 4. Normally it is reasonable to use a two-layer GNN, but the result is very bad when doing so (see Soft-DHN 2 layers result in Sec. A.4.1). Also, the result of the proposed model is still good when not applying a hypergraph GNN. So I'm confused about where the performance gain comes from.

Another weakness is the paper writing. Below are a few comments:
[Page 3, Sec. 3.1] By saying "t\neq\Phi", do you mean "t\neq\emptyset"?
[Page 3, Sec. 3.2] Please introduce the notation (M, C) right after it first appears in the third line of Sec. 3.2.
[Page 3, Sec. 3.2] The notation n,m are confusing. Do you mean m=|E|?
[Page 3, Sec. 3.2] I cannot see how Z=h(\mathcal{H},X_V,X_E) maps each vertex to a probability distribution. Do you mean each row of Z is a probability distribution for each vertex?
[Page 4, Sec. 3.2] Please define E_D. Is E_D the same as E_d?
[Page 4, Sec. 3.3] Be explicit by saying "approximate the hypergraph by a suitable graph", what do you mean by "suitable graph"? Does it mean use cliques in place of hyperedges?
[Page 4, Sec. 3.3] t=0,...,\tau-1

Since the writing significantly affects the paper readability, and the core contribution of the paper seems incremental, I will vote for a reject to this paper.

**Experience Assessment:**

I have read many papers in this area.

**Review Assessment: Checking Correctness Of Derivations And Theory:**

I carefully checked the derivations and theory.

**Review Assessment: Checking Correctness Of Experiments:**

I assessed the sensibility of the experiments.

**Review Assessment: Thoroughness In Paper Reading:**

I read the paper thoroughly.

---

> ### Author Response · Authors · 2019-11-15
> **Our response to Reviewer #1**
>
> Thanks for the review.
>
> $\textbf{On the design being fragile:}$
> We have conducted more ablation studies on the layers of our method and the results are shown in table 4 (section A.4.1).
> As we can see, the optimal configuration for our method is 1 DHN layer and 2 GNN layers.
> 2 DHN layers degrade the performance and we believe this is because of the oversmoothing issue [Li et al., AAAI'18] caused by an additional matrix multiplication by the incidence matrix.
>
>
>
> $\textbf{On an improved writing:}$
> Thanks for the comments on presentation. We have corrected all the suggested changes in the revised version. We have also explicitly said what `"suitable graph” means (clique expansion for HGNN and mediator-based Laplacian for HyperGCN).
>
>
>
> [Li et al., AAAI'18] Deeper Insights into Graph Convolutional Networks for Semi-Supervised Learning

---

### Official Review · AnonReviewer2 · 2019-10-28
**Official Blind Review #2**

**Rating:** 3

**Review:**

In this paper, the authors propose a soft semi-supervised learning approach on a hypergraph. On the one hand, the vertex labels should be not only numerical or categorical variables but also probability distributions. On the other hand, hypergraphs provide a much more flexible mean to encode the real-world complicated relationship compared to the essential pairwise association. Specifically, the authors carefully obtain the generalization error bounds for a one-layer graph neural network. The experiments support the theory and provide the empirical verification of the proposed method. The appendix offers plenty of details that are helpful for the reader to understand both the theoretical and practical perspectives of this paper. Also, the code looks good to me. The structure of the provided codebase is clear and well documented.

I have some questions and suggestions for the authors:
1. In the method section 3.3, the authors said that “A key idea of our approach is to treat each hyperedge as a vertex of the graph” After this transformation, the graph could be super sparse. So I’d like to know more about the efficiency of the proposed method. Because the datasets in this paper could not be considered as a giant graph. The efficiency could be much more critical for large-scale applications. Besides, such conversion could be one of the most significant technical novelty in this paper, which makes me worry about the methodology contribution of this submission.

2. In the theoretical analysis section 4, there are lots of references for the existing lemmas and theorems. It could be much better if the authors could briefly review and summarize these equations before applying them. By the way, I appreciate the detailed appendix at the same time. But such a summary could complete the paper in a more self-contained way. Also, the authors should improve the writing of paper at the same time.

3. In the experiments section 5, I wonder if the authors would consider some more challenging datasets with the larger graph to evaluate the proposed method. Also, the theoretical analysis is about one-layer networks, which looks technical sound to me. However, in practice, we could not only use the one-layer network for graph classification, even for a small graph.

**Experience Assessment:**

I have read many papers in this area.

**Review Assessment: Checking Correctness Of Derivations And Theory:**

I assessed the sensibility of the derivations and theory.

**Review Assessment: Checking Correctness Of Experiments:**

I assessed the sensibility of the experiments.

**Review Assessment: Thoroughness In Paper Reading:**

I read the paper at least twice and used my best judgement in assessing the paper.

---

> ### Author Response · Authors · 2019-11-15
> **Our response to Reviewer #2**
>
> Thanks for the review.
>
> $\textbf{On the efficiency of the method after the transformation:}$
> The time complexity of GNN after the transformation is linear in the number of directed hyperedges. The simple GCN formulation [Wu et al., ICML’19] enables us to efficiently apply our method to large-scale applications.
>
>
>
> $\textbf{On a larger challenging dataset:}$
> Following the reviewer’s suggestion, we experimented on the large-scale arXiv dataset [Clement et al]. The results are shown in Table 2 of our revised paper. As we can see, our proposed soft-DHN outperforms the baselines because it exploits directions among hyperedges (while the baselines do not).
>
>
>
> $\textbf{On the theoretical analysis:}$:
> Our analysis can be applied for a deeper network (with depth $d$) by considering the graph corresponding to the adjacency $B = A^d$ where $A$ is the adjacency of the given graph. This is again motivated by the Simple GCN formulation in which powers of A are used to define graph convolution. Also, as suggested by the reviewer, we have briefly reviewed and summarised the equations in section 4 of our revised paper and improved the writing.
>
>
> [Wu et al., ICML’19] Simplifying Graph Convolutional Networks
> [Clement et al.] On the Use of ArXiv as a Dataset, ICLR 2019 workshop RLGM

---

### Author Response · Authors · 2019-11-15
**Summary of the Rebuttal**

We thank all the reviewers for their reviews.
All the reviewers expressed concerns on the presentation (paper writing). We have addressed the concerns and uploaded a revised version of our submission. We give a summary of our rebuttal below.


$\textbf{Reviewers #2 and #3  suggested evaluation on additional datasets}.$ We have evaluated our proposed method on ACM and arXiv datasets in Table 2 of our paper. The table demonstrates the superiority of our proposed method on the additional datasets as well.


$\textbf{Reviewer #2 wanted to know the efficiency of a key step in our method}.$ The time complexity of the key step in our method is linear in the number of directed hyperedges. We use Simple GCN to efficiently perform the graph convolution operation. Emprically, we have also shown superior performance on the large-scale arXiv dataset


$\textbf{Reviewer #2 had questions on the applicability of our theoretical analysis to multiple layers}.$ Our analysis can be trivially applied to multiple layers ($d$ hops) by considering the graph corresponding to the adjacency $\mathcal{A}=A^d$ where $A$ is the adjacency of the input graph. This is again motivated by the Simple GCN formulation in which powers of the adjacency are used to define the graph convolution.


$\textbf{Reviewer #1 expressed a concern on the model being fragile}.$. To address this concern, we have conducted more ablation studies. The results are shown in table 4 (section A.4.1).  It clearly shows that $1$ layer of DHN and $2$ layers of GNN give the best performance.


$\textbf{Reviewer #3 had concerns on the novelty of the paper}.$ To the best of our knowledge, we are the first to explore GNNs for soft semi-supervised learning. We propose a novel method for directed hypergraphs, demonstrate its improved performance, and provide five benchmark directed hypergraph datasets for soft semi-supervised learning. On the theoretical front, we modified the “gradient” in the Wasserstein space to satisfy the Lipschitz condition required in the algorithmic stability framework and this is not seen in existing literature.

---

### Decision · Program_Chairs · 2019-12-19

**Decision:**

Reject

**Comment:**

This paper proposes and evaluates using graph convolutional networks for semi-supervised learning of probability distributions (histograms). The paper was reviewed by three experts, all of whom gave a Weak Reject rating. The reviewers acknowledged the strengths of the paper, but also had several important concerns including quality of writing and significance of the contribution, in addition to several more specific technical questions. The authors submitted a response that addressed these concerns to some extent. However, in post-rebuttal discussions, the reviewers chose not to change their ratings, feeling that quality of writing still needed to be improved and that overall a significant revision and another round of peer review would be needed. In light of these reviews, we are not able to recommend accepting the paper, but hope the authors will find the suggestions of the reviewers helpful in preparing a revision for another venue.